# Bias in product availability estimates from contraceptive outlet surveys: Evidence from the Consumer's Market for Family Planning (CM4FP) study

**Brett Keller**[1], **Dale Rhoda**[2], **Caitlin Clary**[2], **Claire Rothschild**[3], **Mark Conlon**[4], **Paul Bouanchaud**[5]*, **CM4FP Group**[¶]

**1** Population Services International, Nairobi, Kenya, **2** Biostat Global Consulting, Worthington, Ohio, United States of America, **3** Population Services International, Seattle, Washington, United States of America, **4** Population Services International, Washington, DC, United States of America, **5** Population Services International, London, United Kingdom

¶ Membership of the CM4FP Group is provided in the Acknowledgments.
* pbouanchaud@psi.org

**Data Availability Statement:** CM4FP public datasets exclude all latitude/longitude geographic coordinates because in the full data sets, outlets

## Abstract

Area-based sampling approaches designed to capture pharmacies, drug shops, and other non-facility service delivery outlets are critical for accurately measuring the contraceptive service environment in contexts of increasing de-medicalization of contraceptive commodities and services. Evidence from other disciplines has demonstrated area-based estimates may be biased if there is spatial heterogeneity in product distribution, but this bias has not yet been assessed in the context of contraceptive supply estimates. The Consumer's Marker for Family Planning (CM4FP) study conducted censuses and product audits of contraceptive outlets across 12 study sites and 2–3 rounds of quarterly data collection in Kenya, Nigeria, and Uganda. We assessed bias in estimates of contraceptive product availability by comparing estimates from simulations of area-based sampling approaches with census counts among all audited facilities for each study site and round of data collection. We found evidence of bias in estimates of contraceptive availability generated from simulated area-based sampling. Within specific study sites and rounds, we observed biased sampling estimates for several but not all contraceptive method types, with bias more likely to occur in sites with heterogeneity in both spatial distribution of outlets and product availability within outlets. In simulations varying size of enumeration areas (EA) and number of outlets sampled per EA, we demonstrated that the likelihood of substantial bias decreases as EA size decreases and as the number of outlets sampled per EA increases. Straightforward approaches such as increasing sample size per EA or applying statistical weights may be used to reduce area-based sampling bias, indicating a pragmatic way forward to improve estimates where design-based sampling is infeasible. Such approaches should be considered in development of improved methods for area-based estimates of contraceptive supply-side environments.

can be linked directly to households, meaning that GPS coordinates for outlets would potentially allow identification of households. CM4FP has instead disclosed time/distance matrices to allow partial replication of geographic analyses. See https://www.psi.org/wp-content/uploads/2021/09/CM4FP-GIS-Methodology.pdf for more details. Data from the CM4FP study are available through the Dataverse repository, using the following links: Kenya: Outlet: https://doi.org/10.7910/DVN/DJXKVA; Household: https://doi.org/10.7910/DVN/TQ1MSE; Time/distance matrices: https://doi.org/10.7910/DVN/AP9XGL Nigeria: Outlet: https://doi.org/10.7910/DVN/G31FHL; Household: https://doi.org/10.7910/DVN/IPXDML; Time/distance matrices: https://doi.org/10.7910/DVN/KADZGA Uganda: Outlet: https://doi.org/10.7910/DVN/1NRFSD; Household: https://doi.org/10.7910/DVN/1NRFSD; Time/distance matrices: https://doi.org/10.7910/DVN/OHQ2IZ Further data requests may be sent to the PSI Research Ethics Board, contact: Kelly O'Keefe, Senior Technical Advisor PSI REB kokeefe@psi.org.

**Funding:** Supported by Bill & Melinda Gates Foundation (https://www.gatesfoundation.org/) grant number INV-006705 to Population Services International. BMGF provided input into the study design and provided feedback on a draft of the manuscript.

**Competing interests:** The authors have declared that no competing interests exist.

## Introduction

Outlet surveys of contraceptive availability that use area-based sampling are a common source of estimates for family planning indicators. When availability of family planning commodities at outlets is spatially clustered, and this is not accounted for in area-based sampling strategies, there is potential for biased estimates. Using only data from a survey of outlets that employed a two-stage area-based sampling approach, it is not possible to assess such bias within the survey itself. Outlet census data from the Consumer's Market for Family Planning (CM4FP) study provides an opportunity to explore such bias through simulation approaches, and to test avenues for mitigating it.

Large-scale health service assessments, such as the Demographic and Health Survey's (DHS) Service Provision Assessment and the World Health Organization's Service Availability and Readiness Assessment surveys, are widely used for estimating health systems' readiness to provide contraceptive services [1–3]. Such assessments rely on a master facility list that enumerates public and private health facilities in the country (or geographic region of interest) as a sampling frame, thereby restricting findings to "brick-and-mortar" health facilities [4] and often only to facilities that are formally registered. As a result, supply-side estimates may lack external validity to non-facility outlets; internal validity may also be threatened due to selection bias if unregistered private facilities are less likely to be captured on master facility lists.

With rapid regulatory changes and delivery channel innovation in the field of sexual and reproductive health (SRH), there is growing interest in measuring service delivery outside of health facilities, such as in pharmacies, drug shops, and by informal vendors [5]. This presents challenges to traditional health facility assessment methodologies [4, 6] since these contraceptive outlets are rarely–if ever–captured in a comprehensive list that can be used as a design-based sampling frame.

While previous studies have measured the total contraceptive market in specific localities using censuses [7], thereby precluding the need for area-based sampling, identifying and surveying every service outlet is not a feasible approach for national, multi-year survey programs. The Performance Monitoring for Accountability (PMA) project has addressed this challenge by using a novel, area-based sampling approach to capture information on contraceptive service provision. PMA selects a sample of enumeration areas (EAs) for a household survey and collects data on contraceptive outlets related to those EAs. PMA first conducts a census of all contraceptive outlets located within each EA. This census is used as a sampling frame to select and survey a fixed quota of up to three private and non-facility outlets within each EA. In addition, all public facilities whose catchment area includes the selected EA are included in the sample of public facilities even if the facility itself is outside the boundaries of the EA [8–10]. The PMA "[service delivery point (SDP)] sample thus reflects the services available to a representative population, rather than being representative of all SDPs in the country" [9].

Simple or stratified random sampling of outlets is comprehensive, but requires continuously updated, comprehensive master facility lists (MFLs) of all public and private outlets, which often do not exist [11]. MFLs may not document private outlets as comprehensively as other types of outlets [12]. Such designs are therefore often restricted to public facilities or do not include outlets that are not formal health facilities. Area-based sampling designs such as the approach developed by PMA are thus necessary for measuring the complete contraceptive service environment, particularly in settings with rapid de-medicalization of contraceptive products and care. However, there is relatively little evidence of the validity and precision of estimates of contraceptive availability that are generated from area-based–or spatial–sampling approaches [13]. Area-based sampling approaches, like simple random sampling, are based on the key assumption that sampling units–for example, facilities included in a master facility

list–are independent and identically distributed [14]. When sampling within spatially defined EAs, heterogeneity in the spatial distribution of sampling units within the EA or in the spatial distribution of product availability within those outlets could introduce bias into sampling estimates even if there is no bias in the selection of the EAs. Sampling bias due to spatial heterogeneity has been detected in other disciplines such as environmental health and conservation science [15, 16].

Previous studies have documented spatial heterogeneity in SRH service provision. For example, mapping of facility-based SRH services in four districts in rural Mozambique revealed strong spatial heterogeneity in clinic placement within districts [17, 18], and analysis of health facilities in Nigeria found spatial inequalities between facility placement and population size [19]. However, to our knowledge, spatial-sampling bias has not been described in the context of family planning (FP) provision. Understanding bias in area-based sampling approaches is critical for advancing methodologies that measure service availability and readiness in mixed health systems. Using data from the CM4FP study, a census and longitudinal cohort of all public and private outlets offering contraceptive services within specific geographies in Kenya, Nigeria, and Uganda, we compared estimates of contraceptive product availability generated from area-based sampling simulations to the study's census counts to explore the magnitude of spatial sampling bias and to describe spatial characteristics associated with bias.

## Methods

### CM4FP study design

The Consumer's Market for Family Planning (CM4FP) project was a multi-round longitudinal family planning (FP) outlet census with an accompanying repeated cross-sectional household survey of women aged 18 to 49. The present analysis uses only the FP outlet (supply-side) data. The CM4FP methodology and data are outlined here but have been described in detail elsewhere [20]. The study aimed to test the feasibility and utility of a range of novel and modified approaches to data collection for understanding the supply and the demand sides of the FP markets in predominantly urban and semi-urban sites in Nigeria, Uganda, and Kenya. This study design allowed for directly linking FP users to the outlets where they obtained their most recent FP method. The study also aimed to represent the full FP supply environments to which the sampled consumers had access. CM4FP collected outlet data on a quarterly basis from four sites in each country between 2019 and 2020. The study focused on urban and semi-urban areas to better understand the total FP market, particularly the private sector, in these zones. In each country, four sites were selected from within an urban area of different size (large, medium, small, and semi-urban). In Uganda, there was one site in a rural area instead of in a semi-urban one.

In each site, CM4FP delineated an outer ring, consisting of contiguous administrative wards or parishes to measure the FP supply-side total market. The geographical boundaries for the outer ring at each site encompassed one or more contiguous wards (Kenya and Nigeria) or parishes (Uganda), that were completely censused to measure the total market for FP products and services within each ring-fenced area. To determine the geographic boundary of the outer rings, an initial target area was selected to capture a total of 600 outlets across all sites in each country. The number of outlets included in the study was not statistically predetermined, but instead based on pragmatic considerations allowing for a deep dive into localized family planning markets within budget and time constraints. CM4FP's supply-side outlet census dataset includes outlet and provider characteristics as well as longitudinal data on FP service provision and FP products (including product type, brands, price, availability and current and past stockouts) collected from quarterly product audits in all FP outlets in each study geography.

The CM4FP outlet survey builds on the FPwatch study design, which also conducted an FP outlet census by enumerating, mapping, and surveying all outlets (of all sectors and levels) that offered FP methods and/or services of all types (but excluding outlets that offer only male condoms) [7, 21]. As in FPwatch, CM4FP FP outlets in the study included hospitals, medical centers, clinics, health centers, pharmacies, and drug shops/chemists/Patent and Proprietary Medicine Vendors (PPMVs), but in an extension to FPwatch, CM4FP collected longitudinal data from outlets, returning to censused outlets quarterly. Outlets were eligible for inclusion in the census of FP product and service providers if they had stocked at least one modern FP method (aside from male condoms) or offered any FP services during the past three months. Public and private health provider and health retail outlets of all types within the outer ring, including hospitals, health facilities, pharmacies, patent and proprietary medicine vendors (PPMVs), and drug shops, were screened for inclusion. Outlets that served the military but not the general public were excluded, as were general retailers, bars, hotels, and brothels where only condoms are typically available. In the Lagos and Abia sites in Nigeria, a small number of general retailers/supermarkets offered oral contraceptive pills and/or emergency contraceptive pills, so these outlet types were screened and included if eligible. In the overall CM4FP study, some CHWs were included in the outlet census, but in our analyses, all CHWs are excluded as they did not have a specific geographical location. We also focus on the private sector, and so exclude public facilities from our analyses.

Product audits were conducted for all available contraceptive method and brand/formulation combinations offered and in-stock on the date of the survey. Audited contraceptive products included all modern contraceptive methods, such as male and female condoms, oral contraceptive pills (OCP], emergency contraceptive pills (EC), contraceptive injectables, implants, and hormonal and copper intrauterine devices (IUDs).

While CM4FP also included cross-sectional survey data on women from households in a smaller designated survey area (the inner ring), this paper draws only on the CM4FP supply side outlets (collected across the outer ring). In this paper, "CM4FP study site" is used to describe the outer ring. The study collected GPS coordinates for all outlets and households included in the study. To ensure anonymity and protect the privacy of household respondents, all geographic information below county, state, or district level has been removed from the publicly available data. This paper includes graphs that depict the real spatial relationship between outlets, but to keep outlet identities confidential, map directions (north/south and east/west) have been flipped in some cases, and the actual boundary of the CM4FP study site, which combines contiguous administrative units, has been replaced with a pseudo-boundary to prevent visual identification. By obscuring study site boundaries, spatial relationships between outlets and of the availability of FP products at those outlets have been preserved while minimizing the risk of identifying specific outlets, which could identify specific matched households. More information about the methodology used and data available may be found on the project website (www.cm4fp.org).

## Research ethics and participant consent

Ethical approval was provided by the PSI Research Ethics Board (01.2019 and 04.2019), the AMREF Ethics & Scientific Review Committee in Kenya (P615-2019), the National Health Research Ethics Committee of Nigeria (NHREC/01/01/2007-27/05/2019), the Uganda National Council for Science and Technology review board (SS 5041 and SS 5104), and the Mildmay Uganda Research Ethics Committee (1105–2019). Informed consent was obtained from all household and outlet/CHW survey respondents prior to conducting study procedures. To protect the identify of participants, consent was obtained verbally, except in Uganda where consent was written as mandated by the in-country review board. Verbal consent was witnessed and recorded by fieldworkers using electronic data collection devices. The following

protocol deviations occurred, and were reported to relevant ethics boards during the study: modifications to study incentive strategy and timing; outlet and household mapping and sampling modified during implementation; an extension of period of retention of GPS coordinate data beyond planned timeframe in original protocol; changes to study staffing.

### Inclusivity in global research

Additional information regarding the ethical, cultural, and scientific considerations specific to inclusivity in global research is included in the S1 File.

### Creation of pseudo-EAs

Within each study site we created pseudo-EAs for sampling simulations. The study sites were selected from contiguous administrative units and varied substantially in area and estimated population size. If digitized EA boundaries from the national population census had been available to the study team for each study site, they could have been used in the simulation. Because the true EA boundaries were not available for all sites in a digital format, we generated pseudo-EAs of six sizes, each of which was relative to the overall size of the CM4FP study site. The size of the square pseudo-EAs generated is depicted in Fig 1 with the overall study site pseudo-boundary in blue (Fig 1).

### Simulation approach

To test the potential for bias in different sampling approaches, a simulation was employed (Fig 2). The simulation was conducted using data from four sites each in Kenya, Nigeria, and Uganda, three rounds of data collection, six sizes of pseudo-EAs (termed huge, large, medium, small, smaller, and tiny), four sampling schemes (selecting either 3, 6, 9, or 12 outlets per pseudo-EA), and for eight common FP product availability indicators (male condoms, OCP, EC, injectables, implants, and copper IUDs, plus composite indicators of 3 or more and 5 or more methods of any type, including those not among the 8 common products evaluated).

Product availability estimates from surveys were calculated using repeated draws within each site from a frame of geolocated private outlets where product availability was known from each round of CM4FP data collection. The simulation was restricted to private outlets to best approximate the private sector sampling methods of FP outlet surveys such as PMA. Pseudo-EA boundaries were simulated ten times for each combination of study site, study round, and pseudo-EA size, with boundaries jittered in latitude and longitude for each simulation. For each of the ten sets of pseudo-EAs of a given size, four different sampling schemes were simulated (3, 6, 9, or 12 outlets selected per pseudo-EA), and product availability estimates were produced for each contraceptive method. This approach resulted in 240 estimates per product per round per site (6 pseudo-EA sizes x 10 iterations x 4 sampling schemes).

The estimates were compared with the true or census count of private outlet product availability from all CM4FP data for that site and round. Bias estimates were calculated for each study site, round, and contraceptive method by comparing the census count with the mean availability estimate for each of six pseudo-EA sizes and four sampling schemes. The simulation and analysis were conducted with Stata v17.

## Results

### Study site characteristics

Characteristics of the four CM4FP study sites per country are described in Table 1. Each site was located within a portion of a different urban area and in a different county, state, or district

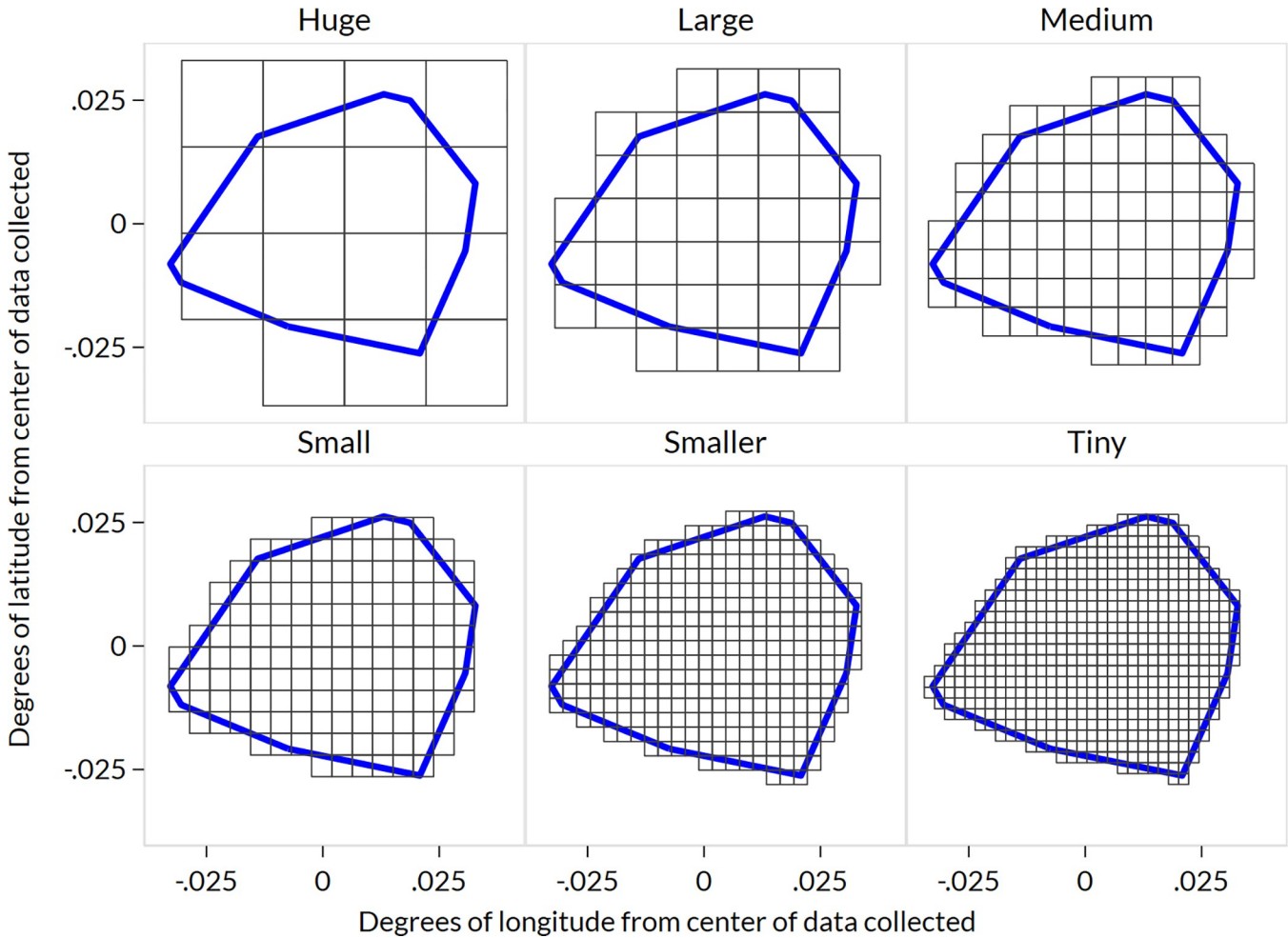

**Fig 1. Pseudo-EA sizes in medium urban site in Kenya.** Pseudo-EAs are depicted as black squares, while the CM4FP study site pseudo-boundary is depicted in blue. In each study site, six pseudo-EA sizes were created, ranging from huge (three entire EAs fit across the study site) to tiny (24 EAs fit across the study site), with the same ratios being used to define these categories of pseudo-EA for each study site. Because the study sites vary in area and population, the geographic footprint and estimated average population of these pseudo-EA sizes also vary across study sites.

(indicated in Table 1). In Kenya, the number of outlets included in the census ranged from 66 to 239 in the semi-urban site and the medium urban site, respectively. In Nigeria the number of outlets surveyed ranged from 86 to 165. In Uganda the number of outlets surveyed ranged from 147 to 157 in three urban study sites, while the rural site had 14 outlets surveyed.

Areas and population estimates for the study sites and corresponding pseudo-EAs are presented in Table 1. Because of differences in study site scale and population density, different sizes of pseudo-EA likely correspond most closely to the true scale and population of census EAs that would be used for sampling of other outlet surveys, such as PMA. We estimated CM4FP study site and pseudo-EA population sizes (Table 1) using data from worldpop.org, with a spatial resolution of 100x100m. [22–24] Further detail on the approach taken is provided in the S2 File.

## Census of FP product availability for comparison with simulation

Fig 3 illustrates the variability of FP product availability within the CM4FP census data (Fig 3). The top panel shows availability for each of four different FP products at the same outlets

1. For each study site (of 12 sites) {

 2. For each study round (of 3 rounds) {

 3. For each pseudo-EA size (of 6 sizes) {

 4. For each of 10 iterations {
 Construct a new set of pseudo-EA boundaries
 5. For each sampling scheme (*n* = 3, 6, 9, or 12 outlets per EA) {

 6. Select up to *n* outlets from each pseudo-EA {

 7. For each product {

$$\text{Estimate: } \textit{Product availability} = \frac{\text{\# of sampled outlets with product available}}{\text{\# of sampled outlets}}$$

 }
 }
 }
 }
 }
 }
}

**Fig 2. Structure of simulation to create pseudo-EAs, sample outlets, and estimate product availability.** This schematic describes the structure of the simulation used to create pseudo-EAs and repeatedly draw samples within them to calculate indicators within each study site, study round, and size of pseudo-EA.

during the same round of data collection, in the Nigeria semi-urban site during round 2. Oral pills were available in 57% of outlets, while implants were available in 6%. The bottom panel of Fig 3 illustrates a different kind of availability, showing that availability of a single product (injectables) varied from 15% of outlets in the small urban site to 33% in the large urban site during round 1 data collection in Nigeria. The spatial pattern of availability varies by product within the same site and round and varies by site within the same product and round.

## Simulation results

Simulation results were summarized using scatter plots called *cone plots* where the y-axis represents number of outlets in the estimate and the x-axis represents product availability, which can range from 0–100% (Fig 4). The census count sits at the apex of the cone, indicated with a red triangle. Each of the four sets of scattered points beneath the apex of the cone represents a different sample size of outlets sampled per pseudo-EA (either 3, 6, 9, or 12).

In cases where the four sets of scatter points are unbiased, they should form a wide-based symmetrical cone with a narrow tip at or near the census count. The decreasing width of the cone as the number of outlets sampled per pseudo-EA increases represents decreasing variance with increased sample size. Scenarios with systematic bias form an asymmetric shape, still wider at the base than at the top, but shaped more like a right triangle where most, or possibly all the mass of the cone falls to the left or right of the census value.

Each Fig 4 cone plot within the top panel represents samples drawn using different sizes of pseudo-EAs. Each cone plot in the bottom panel represents a different product within the same pseudo-EA. If outlets were spaced homogeneously and product availability were also

**Table 1. Characteristics of CM4FP study sites and estimated population size of pseudo-EAs.**

| Country | Study site | CM4FP study site (outer ring) characteristics | | | | | | Median population size estimate for different size pseudo-EAs[c] | | | | | |
| --- | --- | --- | --- | --- | --- | --- | --- | --- | --- | --- | --- | --- | --- |
| | | Number of outlets surveyed [a] | Area (km²) | Outlets per km² | Population size (estimate)[b] | Population density (pop per km²) | Outlets per 10,000 population (estimate) | Huge | Large | Medium | Small | Smaller | Tiny |
| Kenya | Large urban (Nairobi County) | 223 | 14.3 | 15.6 | 281,927 | 19,715 | 7.9 | 47,379 | 11,515 | 6,890 | 3,884 | 1,714 | 919 |
| | Medium urban (Nakuru County) | 239 | 35.6 | 6.7 | 269,572 | 7,572 | 8.9 | 50,635 | 10,263 | 4,393 | 1,888 | 781 | 422 |
| | Small urban (Kilifi County) | 81 | 307.3 | 0.3 | 163,495 | 532 | 5.0 | 21,164 | 4,334 | 1,869 | 967 | 424 | 226 |
| | Semi-urban (Migori County) | 66 | 84.3 | 0.8 | 56,822 | 674 | 11.6 | 4,368 | 1,070 | 460 | 263 | 115 | 64 |
| Nigeria | Large urban (Lagos State) | 150 | 7.3 | 20.5 | 195,204 | 26,740 | 7.7 | 34,880 | 9,395 | 4,155 | 2,327 | 988 | 517 |
| | Medium urban (Kaduna State) | 165 | 22.3 | 7.4 | 205,626 | 9,221 | 8.0 | 24,554 | 7,037 | 3,016 | 1,566 | 719 | 370 |
| | Small urban (Abia State) | 136 | 5.0 | 27.2 | 82,090 | 16,418 | 16.6 | 10,163 | 2,496 | 1,161 | 634 | 323 | 185 |
| | Semi-urban (Niger State) | 86 | 29.4 | 2.9 | 169,241 | 5,756 | 5.1 | 21,438 | 5,862 | 2,336 | 1,237 | 478 | 246 |
| Uganda | Large urban (Kampala District) | 151 | 6.2 | 24.4 | 106,408 | 17,163 | 14.2 | 13,518 | 3,464 | 1,471 | 858 | 334 | 171 |
| | Medium urban (Mbarara District) | 147 | 11.8 | 12.5 | 69,042 | 5,851 | 21.3 | 12,702 | 3,237 | 1,396 | 762 | 325 | 171 |
| | Small urban (Gulu District) | 157 | 33.3 | 4.7 | 153,150 | 4,599 | 10.3 | 32,447 | 7,109 | 2,917 | 1,558 | 644 | 354 |
| | Rural (Soroti District) | 14 | 54.5 | 0.3 | 11,678 | 214 | 12.0 | 2,088 | 480 | 219 | 125 | 53 | 28 |

a. Outlet numbers reported in this table include only static outlets and exclude Community Health Workers, who were included in the CM4FP data collection but who were excluded from the spatial analysis reported in this paper.

b. Population was not directly measured for the CM4FP study site outer ring. Figures were estimated by summing the estimated population in the study site using data from worldpop.org. Estimates are constrained to match 2020 United Nations population estimates at the country level and further constrained to only assign population to settled portions of the countryside.

c. Pseudo-EA sizes are relative to the size of the CM4FP study site and thus not equivalent across sites. Population estimates for pseudo-EAs were generated using the same technique as in footnote b, but applied to the pseudo-EA boundaries, to provide an approximate measure of EA population size for comparison with other sampling approaches.

spatially homogeneous, then the cone plots would all be symmetric. When outlets and the products are distributed with heterogeneous density, the sample estimates can exhibit systematic bias–either biased too high or too low. When bias in the simulated estimate is greater than 5% (a threshold chosen arbitrarily to represent a non-negligible level of bias), it is denoted by a

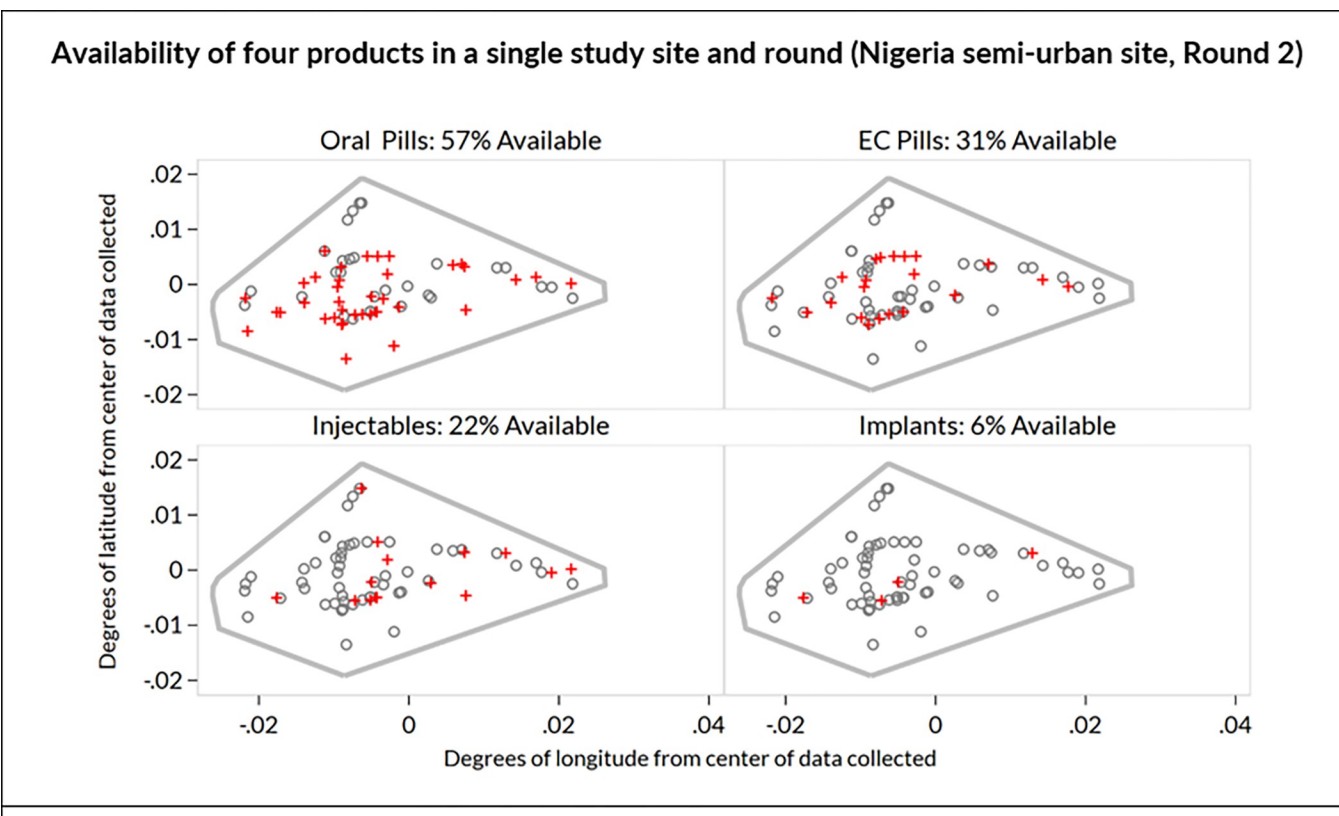

Availability of four products in a single study site and round (Nigeria semi-urban site, Round 2)

Availability of a single product (injectables) across four study sites in one round (Nigeria, Round 1)

+ Available    ○ Not available

**Fig 3. Variability in contraceptive product availability, by product type and by study site.** These maps show which outlets within the CM4FP study site have specific FP products, from a census of all outlets with contraceptive products available beyond just male condoms. Grey circles represent outlets without a product, red pluses represent outlets with a product, and the grey boundary denotes the CM4FP study site pseudo-boundary. The top panel illustrates spatial heterogeneity of availability varying across products. The bottom panel illustrates spatial heterogeneity of availability varying across the same product in different study sites.

downward-facing arrow (V) at the top of the cone plot where the point of the triangle aligns with the mean simulated proportion.

The top panel of Fig 4 plots 240 simulated availability estimates for male condoms in the Kenya small urban site, round 2. All six cone plots exhibit notable bias, where the sample-based estimates are more likely to be below the census figure than above it. When the pseudo-EAs are huge, the number of outlets in each sample is relatively small, so the variance is large. As pseudo-EA size decreases, the number of pseudo-EAs increases so the number of outlets per sample increases and the variance diminishes but the sample bias remains. Even when the pseudo-EA size is tiny, most estimates fall below the census figure, but the average bias is still greater than 5% when only 3 outlets are sampled per EA.

The bottom panel of Fig 4 plots 40 availability estimates for each of six different products in the Kenya semi-urban site, round 1, holding the pseudo-EA size fixed at large (meaning that about six entire pseudo-EAs would fit across the CM4FP study site). In this panel, the simulated estimates exhibit bias for the three products in the top row (stocking three or more methods, male condoms, and OCPs), and those in the bottom row (stocking five or more methods, injectable, and implant) do not. For some product availability estimates to be biased and others to not be biased means that the bias is not only a function of the density or distribution of outlets but must also be related to spatial heterogeneity of the products across the outlets.

We explored the nature of the heterogeneities of outlet distribution and product availability that may result in bias in simulated samples by plotting a histogram, line graph, and cone plot for a specific CM4FP study site (Fig 5); each are explained in turn below.

Spatial concentration of outlets at each site was examined using a histogram of the number of other outlets located within 2km of each outlet. Perfect homogeneity would be represented by a single tall bar in the histogram (e.g., every outlet has 10 other outlets within 2km). Heterogeneity is characterized by spread in the histogram and by heavy right or left-side tails in the distribution (i.e., where many outlets have many other outlets within 2 km and many outlets have very few within 2 km) (Fig 5).

Even with widely varying densities of outlets, it would be possible to have a homogenously available product (e.g., for every outlet, 70% of the outlets within 2km carry the product). To characterize heterogeneity of product availability overlaid on outlet concentration, we plotted the percentage of outlets within 2 km of each outlet that have the product available versus outlet concentration in line graphs (Fig 5). Blue dots represent observed values for each outlet with a blue line of best fit. The horizontal red line indicates the availability of the FP product at the census proportion count, assuming a homogenous spatial distribution. If product availability is perfectly homogeneous, the blue best-fit line will be horizontal and lie atop the red line at the y-values of the census measure of product availability. But if product concentration varies with outlet concentration, then the blue best-fit line will have a non-zero slope and cross the red census availability line. A non-zero slope in a blue line is a reliable predictor of bias and tends to correspond with a skewed cone plot even though the concentration heterogeneity plot is calculated from a census of outlets and the bias is demonstrated to be a sampling phenomenon. The non-zero slope in the census-based plot is an indicator of an environment that may yield biased samples.

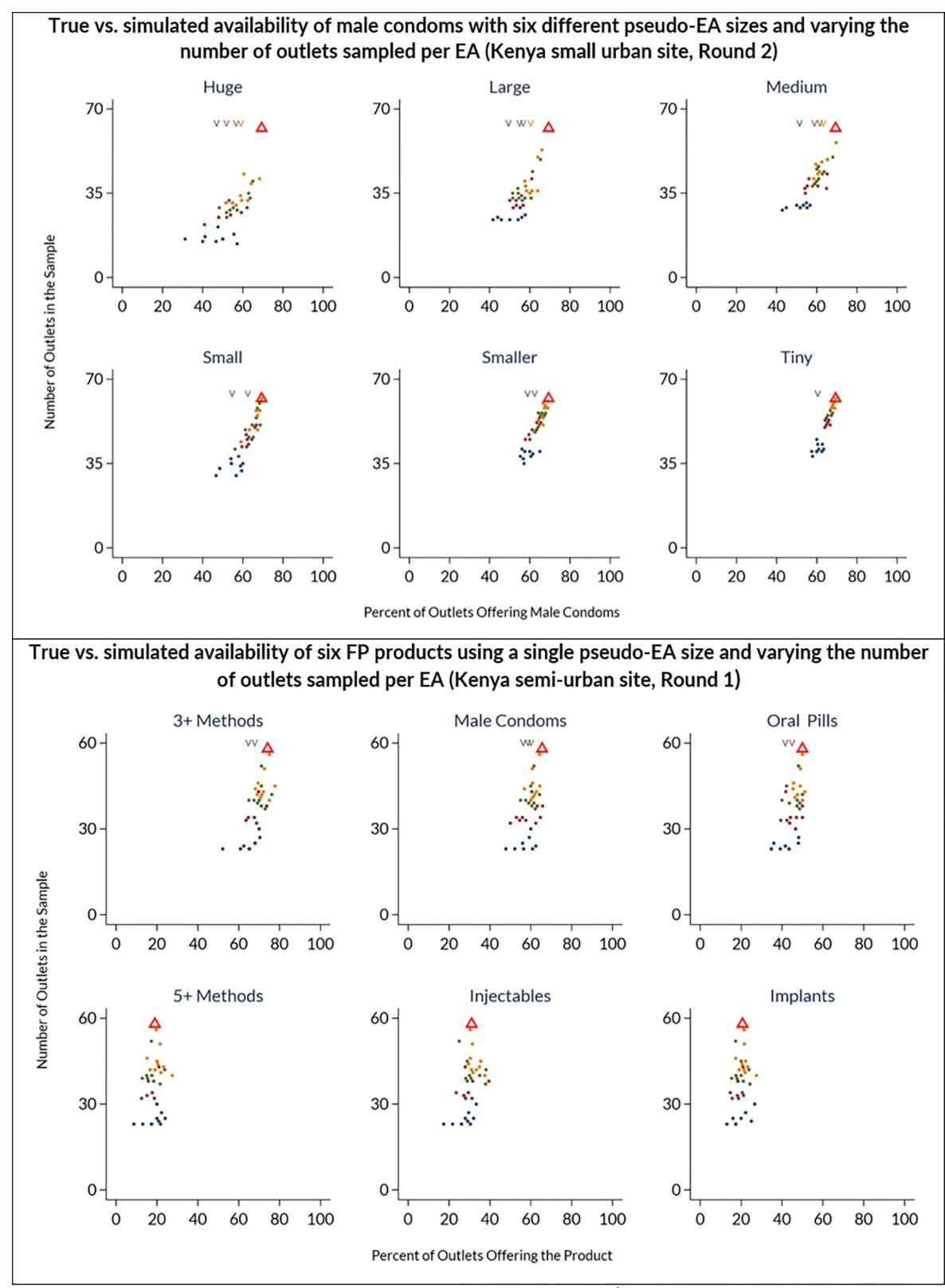

**Fig 4. Simulated sampling bias cone plots.** Each scatter plot depicts the census (true) availability measure as a red triangle, along with point estimates of availability from simulated samples that drew 3, 6, 9, and 12 outlets per EA. Simulations with varying sample size per EA are arranged from smallest sample (n = 3 per EA) in blue at bottom of cone to largest sample (n = 12 per EA) in red at top of cone. In scenarios where simulated samples show systematic bias, the points form an asymmetric shape. When bias of the simulated metric exceeds 5%, a downward-facing arrow denotes the simulated average at the top of the plot. This illustrates a general finding from the simulations: bias often occurred for some, but not all, indicators.

Fig 6 demonstrates how the mean bias of groups of ten simulated samples varies by pseudo-EA size, by sample size per pseudo-EA, and by heterogeneity of outlet density (Fig 6). Red points in the figure indicate that a batch of ten samples had a mean bias with absolute value ≥ 5%. Note that bias is most common when pseudo-EA size is large and when few outlets are selected per pseudo-EA–represented in the sub-plots near the upper left corner of the figure–both factors that keep the simulated sample size small. Within sub-plots, bias is more likely to be positive at the left side of the plots, where distributions of outlet concentration do not have heavy tails. Conversely, bias is more likely to be negative at the far-right side of the sub-plots, in particular in locations where outlet concentration varies substantially (meaning that a large portion of outlets appear in the outer two-thirds of the concentration distribution).

Our simulation demonstrated that bias is more likely to occur with Eas that are larger; bias greater than 5% (positive or negative) occurred in 30% of simulations in huge pseudo-Eas, and just 2% of simulations in tiny pseudo-Eas.

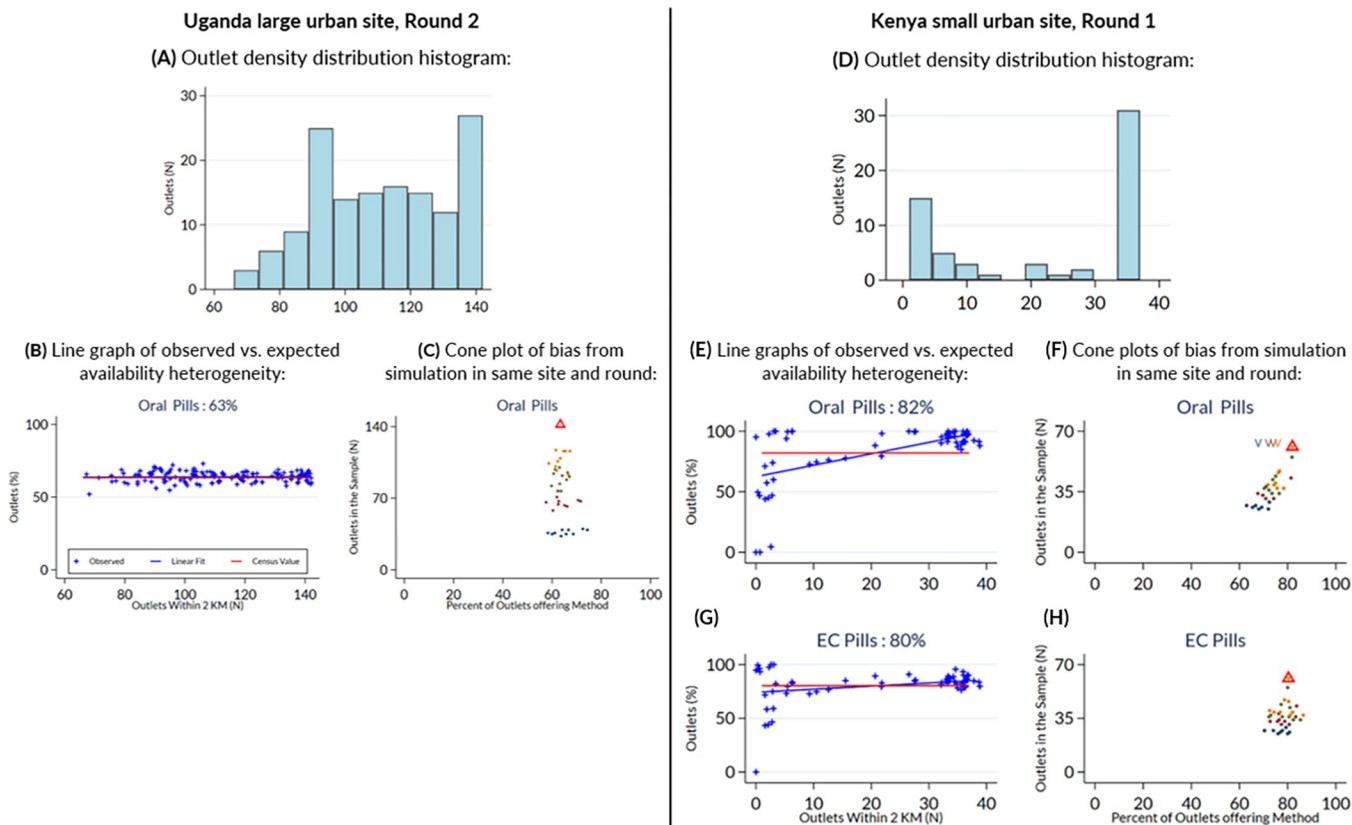

**Fig 5. Bias as a function of outlet concentration and heterogeneity in product availability.** In the left panel, outlet density is more constant than in many distributions (**A**), but the availability of OCP does not vary with density (**B**), and sampling bias is therefore low (**C**). In the right panel, outlet density varies more starkly, producing a clearly bimodal distribution (**D**). When product availability is heterogeneous (**E**), bias is seen in the simulation cone plot (**F**). When product availability is homogenous (**G**), bias is not seen in the simulation cone plot (**H**). Bias in the simulated results requires both outlet density heterogeneity and product availability heterogeneity.

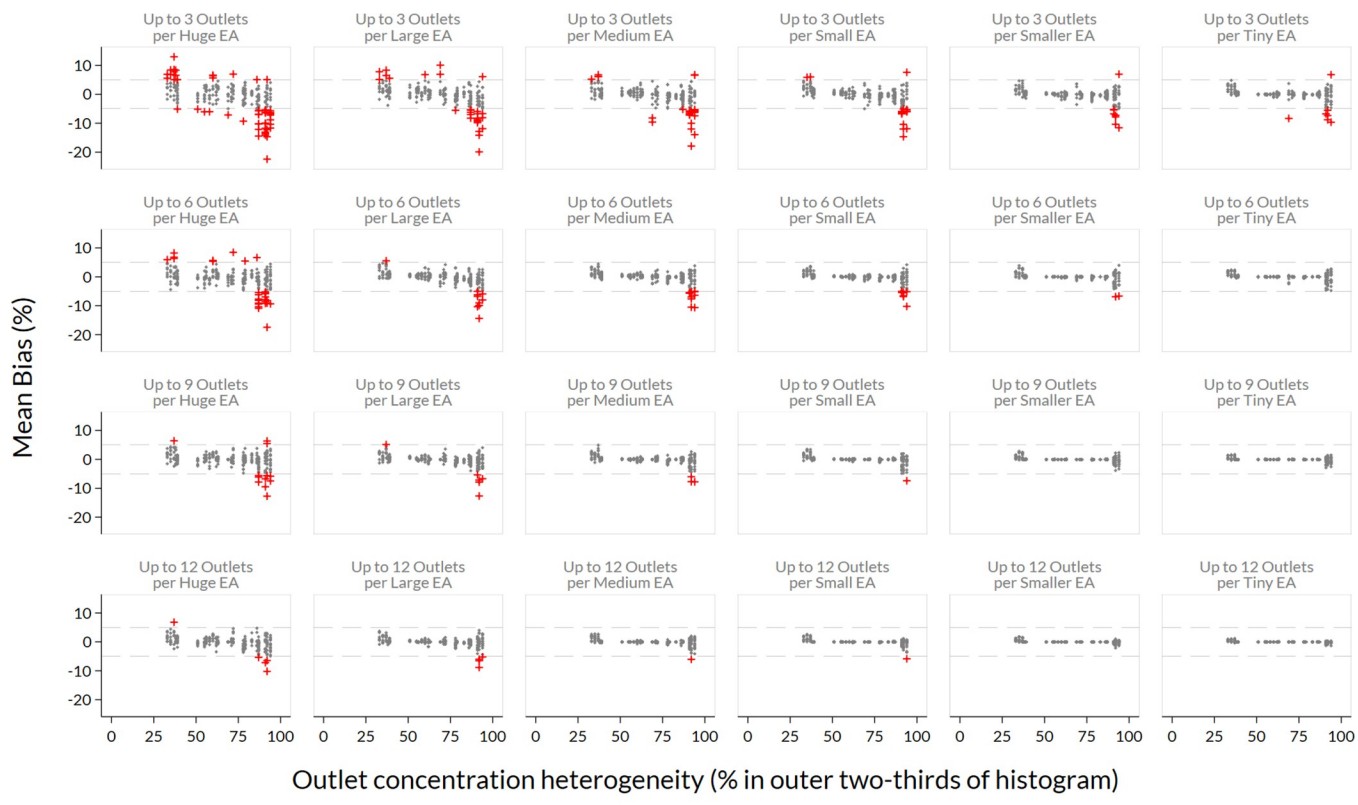

**Fig 6. Sampling bias by EA size, sample size, and heterogeneity of outlet density.** When outlet concentration is more heterogeneous across the study site, estimates tend to be biased negatively; estimates are more likely to be positive biased when outlet concentration heterogeneity is low. In the CM4FP data, product availability tends to be lower in areas where outlets are less dense, producing this pattern. Notable bias is more often observed when EAs are larger, and the number of outlets sampled per pseudo-EA is smaller. This figure pools data from simulations of all product availability indicators and data from all 12 CM4FP study sites and all rounds of data collection.

In our simulations, bias of greater than 5% occurred in 20% of simulations with 3 outlets sampled per pseudo-EA, 10% of simulations with n = 6, 6% of simulations with n = 9, and 4% of simulations with n = 12. Fig 7 summarizes the proportion of simulations where bias greater than 5% from the census count was observed, by country, method, and outlets sampled per pseudo-EA (Fig 7). Bias was most common in estimates of male condoms and OCP, and generally more common in the Kenya data. Estimates were more likely to be biased downward than upward. Across all three countries and all availability indicators, bias decreased with larger numbers of outlets sampled per pseudo-EA.

We tested a simple statistical weighting approach using the same simulation approach. For each simulated sample, the outlets selected from a pseudo-EA were assigned a survey weight equal to the total number of outlets in the pseudo-EA, such that outlets selected from pseudo-Eas with a larger total number of outlets are more influential for summary estimates than outlets selected from pseudo-Eas with a smaller total number of outlets. Fig 8 illustrates a reduction in bias when this weighting technique was applied to one of the most biased estimates, that for male condom availability in the small urban site in Kenya, in CM4FP data collection round 2 (Fig 8).

| | Kenya | | | Nigeria | | | Uganda | | |
|---|---|---|---|---|---|---|---|---|---|
| | Proportion of estimates with greater than 5% negative bias | Proportion of estimates with greater than 5% positive bias | Mean absolute bias (%) | Proportion of estimates with greater than 5% negative bias | Proportion of estimates with greater than 5% positive bias | Mean absolute bias (%) | Proportion of estimates with greater than 5% negative bias | Proportion of estimates with greater than 5% positive bias | Mean absolute bias (%) |
| **3+ Methods** | | | | | | | | | |
| 3 outlets per EA | 17% | 11% | 3.9 | 11% | 6% | 1.5 | 12% | 11% | 1.9 |
| 6 | 8% | 5% | 2.2 | 8% | 3% | 1.0 | 3% | 3% | 0.9 |
| 9 | 3% | 4% | 1.5 | 3% | 1% | 0.5 | 2% | 0% | 0.4 |
| 12 | 3% | 2% | 1.0 | 3% | 1% | 0.5 | 1% | 0% | 0.4 |
| **5+ Methods** | | | | | | | | | |
| 3 outlets per EA | 11% | 14% | 2.0 | 2% | 2% | 0.5 | 2% | 2% | 0.5 |
| 6 | 4% | 7% | 1.3 | 2% | 1% | 0.4 | 1% | 1% | 0.3 |
| 9 | 2% | 3% | 1.0 | 0% | 0% | 0.2 | 0% | 2% | 0.3 |
| 12 | 2% | 4% | 0.8 | 0% | 0% | 0.1 | 0% | 0% | 0.1 |
| **Copper IUDs** | | | | | | | | | |
| 3 outlets per EA | 11% | 8% | 1.5 | 3% | 5% | 1.1 | 2% | 3% | 0.6 |
| 6 | 3% | 3% | 0.6 | 2% | 0% | 0.7 | 0% | 0% | 0.3 |
| 9 | 1% | 2% | 0.5 | 0% | 0% | 0.3 | 0% | 0% | 0.2 |
| 12 | 1% | 3% | 0.5 | 0% | 0% | 0.2 | 0% | 0% | 0.1 |
| **EC Pills** | | | | | | | | | |
| 3 outlets per EA | 22% | 7% | 2.9 | 13% | 6% | 1.4 | 18% | 8% | 2.4 |
| 6 | 14% | 3% | 2.0 | 4% | 4% | 0.9 | 12% | 3% | 1.0 |
| 9 | 10% | 4% | 1.5 | 4% | 1% | 0.5 | 1% | 2% | 0.4 |
| 12 | 6% | 2% | 0.9 | 2% | 1% | 0.5 | 2% | 1% | 0.4 |
| **Implants** | | | | | | | | | |
| 3 outlets per EA | 13% | 11% | 1.9 | 1% | 5% | 1.0 | 3% | 2% | 0.7 |
| 6 | 4% | 7% | 1.4 | 1% | 1% | 0.5 | 1% | 0% | 0.3 |
| 9 | 3% | 3% | 1.1 | 0% | 0% | 0.2 | 0% | 0% | 0.2 |
| 12 | 1% | 3% | 0.8 | 0% | 0% | 0.2 | 0% | 0% | 0.1 |
| **Injectables** | | | | | | | | | |
| 3 outlets per EA | 9% | 22% | 2.7 | 14% | 4% | 1.8 | 10% | 12% | 1.6 |
| 6 | 2% | 9% | 1.7 | 8% | 4% | 1.2 | 11% | 2% | 1.0 |
| 9 | 2% | 6% | 1.0 | 2% | 2% | 0.5 | 2% | 2% | 0.4 |
| 12 | 0% | 8% | 1.1 | 3% | 1% | 0.4 | 0% | 0% | 0.2 |
| **Male Condoms** | | | | | | | | | |
| 3 outlets per EA | 44% | 5% | 5.0 | 8% | 8% | 1.7 | 14% | 11% | 1.5 |
| 6 | 30% | 3% | 3.3 | 3% | 5% | 0.9 | 10% | 1% | 0.8 |
| 9 | 20% | 2% | 2.5 | 3% | 3% | 0.5 | 1% | 1% | 0.3 |
| 12 | 12% | 1% | 1.7 | 2% | 2% | 0.4 | 1% | 1% | 0.3 |
| **Oral Pills** | | | | | | | | | |
| 3 outlets per EA | 41% | 7% | 5.5 | 10% | 8% | 1.2 | 10% | 5% | 2.4 |
| 6 | 29% | 6% | 3.5 | 4% | 3% | 0.7 | 3% | 2% | 0.9 |
| 9 | 15% | 3% | 2.2 | 2% | 1% | 0.4 | 2% | 0% | 0.6 |
| 12 | 11% | 1% | 1.4 | 1% | 0% | 0.3 | 0% | 0% | 0.4 |

Note: Data bars are scaled so that a value of 50% would fill the entire width of the cell.

**Fig 7. Proportion of round 1 simulated product availability estimates with bias of greater than 5%.** The proportion of simulated estimates that were biased negatively by 5% or more and positively by 5%, as well as the mean absolute bias, is presented by country for each of the product availability indicators and for

four different sampling scenarios (3,6, 9, or 12 outlets per pseudo-EA). Bias greater than 5% in either direction, and mean absolute bias, tend to decrease as sample size per pseudo-EA is increased, with the largest reduction in bias occuring when sample size increased from 3 to 6 outlets per pseudo-EA.

## Discussion

We found evidence of potential bias in FP outlet surveys by simulating different sampling schemas and comparing the resulting estimates of FP product availability with the census means. In some cases, we found substantial levels of bias in product availability estimates. We found that heterogeneity in outlet spatial distribution and in product availability were associated with greater likelihood of bias. Differential bias by FP product type may be explained by the differential levels of availability of FP products within each outlet, and differential distributions of FP products across outlets in a geographic area. The following section explores potential mechanisms for the bias seen in the results, before discussing the implications of these findings.

### Understanding how bias might arise

We found evidence that bias in indicator estimates based on samples may arise when there is heterogeneity in outlet distribution in the wider areas from which the EA is drawn. This bias may under- or over-estimate the indicator of interest (as compared to its census value).

Notionally, consider two types of Eas within a single study site: outlet-sparse Eas with a few outlets that are all selected into every sample and outlet-dense Eas with so many outlets that different samples include different subsets of outlets. If the average availability of products is substantially higher in outlet-dense Eas than in sparse Eas, then every sample of the entire study site will underestimate the true proportion of outlets that carry the product; the cone plots will be skewed with all sample points falling below (to the left of) the census value. This bias will persist in estimates for a larger region. Conversely, if the product were more available in outlet-sparse Eas than outlet-dense ones, then the samples would systematically overestimate availability. The principle holds even with a continuum of outlet density if there is a correlation between outlet density and product availability.

Fig 9 shows a simplified illustration of how this type of bias may arise, depicting a scenario in which product availability varies at outlets across three hypothetical Eas (Fig 9). When outlets are sampled using area-based sampling under these conditions, estimates are usually biased in comparison to the census results.

### Implications for outlet surveys that use sampling approaches

Identifying bias in estimates generated from area-based sampling approaches is critical for several reasons. First, if estimates are used for situational analyses or assessment of trends in FP delivery and availability, understanding the magnitude of potential bias and whether bias is differential with respect to characteristics of the sampled areas is critical for interpretation of findings. Differential bias is particularly problematic if it results inappropriate resource allocation or planning due, for example, to systematic underestimation or overestimation of contraceptive availability in certain areas. Because the bias described in this paper can differ for different products within the same CM4FP study site, it can also lead to incorrect conclusions about the relative availability of one product compared to others.

Second, biased area-based sampling estimates would be expected to bias estimated associations between contraceptive supply and demand. Estimating relationships between contraceptive services and individual contraceptive behaviors is an active area of research that attempts to link household-based health surveys with the nearest or nearby health facility assessments

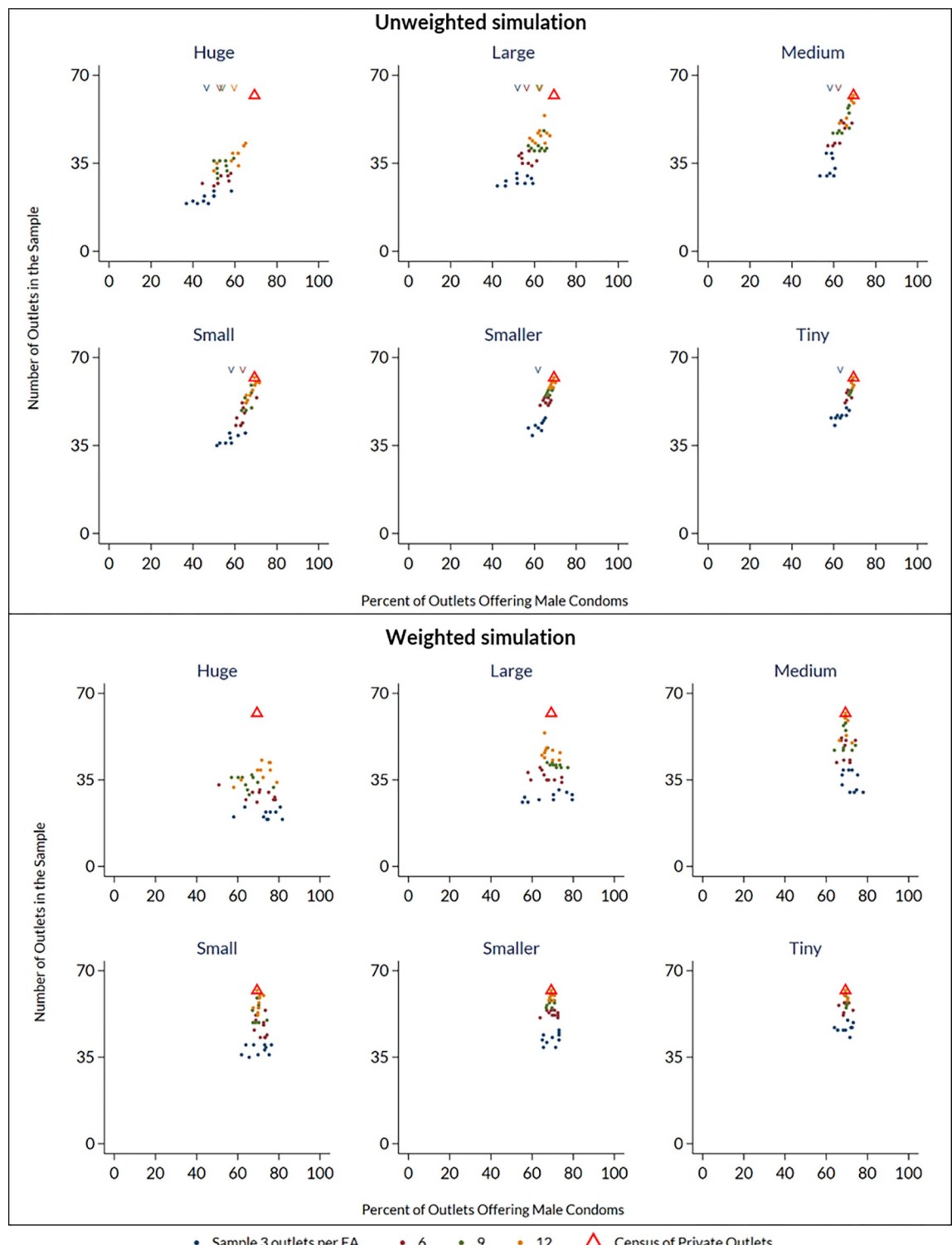

**Fig 8. Effect of applying statistical weights in reducing bias in simulated estimates of product availability.** As in Fig 4, each scatter cone plot depicts the census (true) availability measure as a red triangle, along with point estimates of availability from simulated samples that drew 3, 6, 9, and 12 outlets per EA. Simulations vary by sample size per EA and by relative size of pseudo-EA (huge, large, medium, small, smaller, and tiny). In scenarios where simulated samples show systematic bias, the points form an asymmetric shape. When bias of the simulated metric exceeds 5%, a downward-facing arrow denotes the simulated average at the top of the plot. The top

panel shows simulation results without statistical weighting, indicating substantial bias is present in most scenarios. The bottom panel shows simulation results in which selected outlets are weighted based on the number of total number of outlets in the pseudo-EA, indicating the weighting approach alleviates bias.

[13, 25–27]. Existing evidence is mixed, and difficulties accurately measuring the total FP market–particularly in urban settings with active private and non-facility FP markets–have been cited as a potential explanation for these inconsistent associations [25]. Improved approaches to sampling from total FP markets that provide unbiased estimates of the local FP market (including private sector providers and non-facility outlets) will be critical for better understanding how supply-side factors influence contraceptive intentions and behaviors, including method uptake, switching, continuation, and satisfaction.

## Limitations

The CM4FP study on which our simulation is based had an urban focus, meaning that conclusions may not be readily generalized to rural contexts, (though the study does include one rural site, in Soroti District, Uganda), where private sector facilities and outlets are less prevalent. In theory, such biases may be present in rural outlet surveys, but we were unable to directly test this hypothesis. Additionally, the CM4FP data came from a limited number of study sites and were not intended to constitute nationally or sub-nationally representative samples. Thus, results evaluated for bias in this paper, such as product availability, should be interpreted as being from 12 distinct study sites spread across three countries, rather than being representative of higher geographical units or levels. Moreover, study sites accounted for a limited geographic area within each setting and results evaluated are not statistically representative of the wider geographical area or urban setting. While not geographically representative, we consider the findings presented in the manuscript to be nevertheless generalizable to other studies that employ sampling approaches similar to those simulated here.

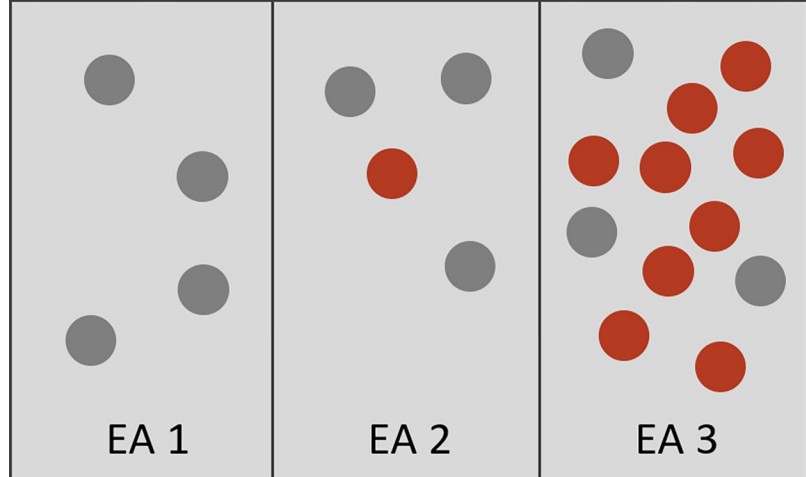

| Outlets sampled per EA | Estimate of mean product availability |
|---|---|
| 1 | 33.3% |
| 3 | 33.3% |
| 6 | 39.3% |
| Census | 50% |

🔴 Product available    ⚫ Product not available

**Fig 9. Stylized illustration of how bias may arise.** In this stylized scenario, there are 20 outlets spread unequally across three enumeration areas. Product availability is 50% across all 20 outlets, but varies from 0% to 75% by EA. If samples of outlets are drawn from the universe of outlets in the EA, mean availability varies but can be consistently biased.

The exclusive focus on the private sector in this analysis is also a limitation. This simulation approach focused on private outlets due to their larger number within CM4FP study sites and especially because of the close parallel with existing area-based sampling of outlets, such as in PMA. Because the PMA sampling approach for public facilities differs, the findings of our simulated approach cannot be applied directly to public sector PMA results, and do not describe the localized total market. For outlet surveys where public sector sampling parallels PMA's private sector outlet sampling approach, the same biases are possible if public sector spatial heterogeneity and product availability display similar patterns.

Last, the inability to access census Eas across all sites increased the required simulation volume. The sheer volume of simulation output was multiplied six-fold because we did not have census EA boundaries for the study sites and wanted to cover a range of possible EA sizes. It is likely that the census Eas would match different simulation EA sizes at different sites. Fig 6 shows some notable examples of both positive and negative bias in even the smallest Eas when only three outlets are sampled per EA, so although we do not know precisely which EA sizes are most realistic at each site, the potential for bias exists in some of these sites and products at externally valid scales (Fig 6).

## Recommendations for outlet surveys

Other studies that employ sampling approaches like those simulated in our analysis might wish to consider the risk of bias that may potentially result. Possible remedies include varying sampling approaches when spatial heterogeneity is known to exist, increasing sample size, and weighting data. In the common case where such *a priori* data do not exist, study designers will need to choose between using limited resources to increase sample size within Eas and tolerating potential bias in estimates. In addition to increasing sample size per EA where possible, survey designers should consider planning for statistical weighting approaches that may reduce such bias regardless of sample size. We explore potential options for limiting bias in more detail below.

The literature on spatial sampling suggests several techniques that may remedy bias, though some of these techniques may not be operationally efficient. If product availability variance were known *a priori*, then the number of outlets to be sampled per EA could be "assigned to each subarea according to the area and/or variance proportion," but this is unlikely to be practical [14]. Another solution suggested by the same authors is that spatial statistics may be used to apply corrections to existing datasets if the factors responsible for bias (such as heterogeneity of outlet spacing combined with heterogeneity of product availability) have been identified and are available for a broader dataset of outlets. Finally, sampling bias can be mitigated by creating a sampling frame of outlets within an enumeration area and applying a spatial sampling technique wherein the geographic spacing of outlets is considered rather than simple random sampling. [14]. Notably, these proposed approaches to spatial sampling require specialized sampling techniques based on spatial data from an outlet census, a process that is both cost- and time-intensive.

We found that the likelihood of substantial bias decreases as EA size decreases and as the number of outlets sampled increases. These findings suggest that decreasing EA size may be one approach to minimizing area-based sampling bias. However, relying on existing population census-derived Eas is common for operational reasons, so varying their size in area-based sampling designs may not be feasible. Another simple method for mitigating against this bias would be to sample more outlets per selected EA. The simulation approach we employed indicates that bias originating from spatial characteristics of outlet location and product availability may be reduced–though not eliminated–in the design phase by drawing a sample of more

outlets per EA. For example, in simulations of the availability of 3 or more methods at an outlet, increasing the per-EA outlet sample from 3 to 6 outlets per pseudo-EA decreased the occurrence of bias greater than 5% from 28% of simulations to 13% of simulations. A larger sample size is more important in the presence of spatially heterogeneous outlet distribution and product availability.

It may be possible to mitigate the bias by applying survey weights to availability estimates, by weighting outlets based on the total number of outlets present in the EA or by other criteria. We illustrated this weighting approach in Fig 8. More systematic work to develop optimal weighting techniques for outlet surveys may be required, including weighting approaches that may be relevant for surveys already collected, as well as for planned outlet surveys.

Common area-based outlet sampling approaches may be prone to bias in product availability estimates. When certain conditions of outlet spatial heterogeneity and product availability coincide, for private outlets in urban settings. It is difficult to assess the presence of such factors without a comprehensive census of all outlets in an area, so we cannot directly estimate the prevalence of such bias in published surveys that do not use a census approach. Instead, surveys of healthcare outlets, including FP outlet surveys, should where possible consider implementing measures to mitigate bias, either at the stage of study design, or retrospectively through weighting.

## Conclusions

Area-based sampling designs are increasingly being used to estimate complex healthcare markets. We used a simulation approach to compare estimates derived from area-based sampling approaches with estimates from a full census of outlets providing FP products and services. We demonstrated that bias in area-based sampling estimates of contraceptive product availability is common and is more likely to occur when spatial heterogeneity in outlet distribution and product availability are both present. In some areas this bias can be large enough to affect conclusions about the relative availability of different products and could potentially lead to poor policy or programmatic decision-making. Other researchers designing studies to measure FP supply-side indicators through outlet surveys should consider the tradeoff between risk of bias, and the increased resources needed to increase sample sizes or–more likely to be cost-effective–to make other sampling design and analysis changes to limit that risk. We demonstrate that simple techniques employed in survey design and data analysis through weighting may mitigate substantial bias arising from area-based sampling in the presence of spatial heterogeneity. These approaches may present practical ways forward for improving estimates of service availability where it is impractical or infeasible to use design-based sampling approaches.

## Supporting information

**S1 File. Inclusivity in global research.**
(PDF)

**S2 File. Pseudo-EA population estimation.** A brief description of population estimates for pseudo-enumeration areas.
(PDF)

## Acknowledgments

We would like to thank the study participants for generously sharing their time and information with this project. We are also grateful to the CM4FP Group and data collection teams,

supervisors, and field staff for their contributions to data collection and analysis between 2019 and 2021.

CM4FP Group 2019–2021: Dr Jennifer Anyanti; Justin Archer; Dr. Kimberly Ashburn; Henry Bakira; Dr Paul Bouanchaud (Technical Director 2021) pbouanchaud@psi.org; Peter Buyungo; Dr Caitlin Clary; Mark Conlon; Eden Demise; Kevin Duff; Dr Uche Ekhator-Mobayode; Hoda Elmasry; Dr Hildah Essendi; Jordan Freeman; Dr Susannah Gibbs; Risa Griffin; Dr Nathan Heard; Dr Bo Hu; Ashley Jackson; Dr Amanda Kalamar (Technical Director 2018–21); Brett Keller; Irene Kyomuhangi; Baker Lukwago; Dr Peter Macharia; Alison Malmqvist; Harmon Momanyi; Micheal Mugerwa; Doreen Nakimuli; Julius Njogu; Dr Anthony Nwala; Noah Nyende; Daniel Olemo; Jacob Olila; Alyssa Om'Iniabohs; Chinedu Onyezobi; Dale Rhoda; Dr Claire Rothschild; Raymond Songo; Raymond Sudoi; Ekerette Emmanuel Udoh; Dr Nkemdiri Wheatley (Principal Investigator 2018–21); Dr Wei Yang

## Author Contributions

**Conceptualization:** Brett Keller, Dale Rhoda, Paul Bouanchaud.

**Data curation:** Mark Conlon.

**Formal analysis:** Dale Rhoda, Caitlin Clary.

**Methodology:** Brett Keller, Dale Rhoda, Mark Conlon, Paul Bouanchaud.

**Project administration:** Mark Conlon.

**Supervision:** Paul Bouanchaud.

**Visualization:** Brett Keller, Dale Rhoda, Caitlin Clary.

**Writing – original draft:** Brett Keller, Dale Rhoda, Claire Rothschild.

**Writing – review & editing:** Mark Conlon, Paul Bouanchaud.

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
