## [Decision Letter · Decision Letter 0]

7 Apr 2022

PONE-D-21-34612Bias in product availability estimates from contraceptive outlet surveys: evidence from the Consumer’s Market for Family Planning (CM4FP) studyPLOS ONE

Dear Dr. Bouanchaud,

Thank you for submitting your manuscript to PLOS ONE. After careful consideration, we feel that it has merit but does not fully meet PLOS ONE’s publication criteria as it currently stands. Therefore, we invite you to submit a revised version of the manuscript that addresses the points raised during the review process.

Authors are advised to elaborate on causes and the impact of different types of bias. In the discussion section, please mention about why only selective contraceptive methods were chosen for biased sampling estimates. Also please elaborate clearly about why likelihood of substantial bias decreases as EA size decreases and when the number of outlets sampled per EA increases. 

We look forward to receiving your revised manuscript.

Kind regards,

Manoj Kumar

Academic Editor

PLOS ONE

Journal Requirements:

3. Please include a complete copy of PLOS’ questionnaire on inclusivity in global research in your revised manuscript. Our policy for research in this area aims to improve transparency in the reporting of research performed outside of researchers’ own country or community. The policy applies to researchers who have travelled to a different country to conduct research, research with Indigenous populations or their lands, and research on cultural artefacts. The questionnaire can also be requested at the journal’s discretion for any other submissions, even if these conditions are not met.  Please find more information on the policy and a link to download a blank copy of the questionnaire here: https://journals.plos.org/plosone/s/best-practices-in-research-reporting. Please upload a completed version of your questionnaire as Supporting Information when you resubmit your manuscript.

5. One of the noted authors is a group or consortium [insert name of group or team]. In addition to naming the author group, please list the individual authors and affiliations within this group in the acknowledgments section of your manuscript. Please also indicate clearly a lead author for this group along with a contact email address.

Reviewers' comments:

Reviewer's Responses to Questions

**Comments to the Author**

1. Is the manuscript technically sound, and do the data support the conclusions?

Reviewer #1: Partly

2. Has the statistical analysis been performed appropriately and rigorously? 

Reviewer #1: Yes

3. Have the authors made all data underlying the findings in their manuscript fully available?

Reviewer #1: No

4. Is the manuscript presented in an intelligible fashion and written in standard English?

Reviewer #1: Yes

5. Review Comments to the Author

Reviewer #1: Review Comments to the Author

The study examined the bias in product availability estimates from contraceptive outlet surveys in Kenya, Nigeria, and Uganda. They found evidence of bias in estimates of contraceptive availability generated from simulated area-based sampling. Within specific study sites and rounds, they observed biased sampling estimates for several but not all contraceptive method types, with bias more likely to occur in sites with heterogeneity in both spatial distributions of outlets and product availability within outlets. I commend the authors for the work done. Authors are encouraged to consider the following issues.

1. The introduction does not highlight the reason for urban focus. Is there a special reason for limiting it urban? Authors could explain the reasons for the selection of geographics in the introduction section.

Can authors also explain bias in product availability estimates from contraceptive outlet in public sectors? It would be good to understand how bias in product availability estimates from contraceptive outlet has been across the countries in the introduction.

There are different causes for bias including area-based bias. Authors should also dedicate a paragraph stating those causes and the impact of different types of bias.

2. The authors stated that for the purpose of the study, they visited hospitals, medical centers, clinics, health centers, pharmacies, and drug shops/chemists/Patent and Proprietary Medicine Vendors (PPMVs), in the study sites. Authors should be clear whether they visited all facilities for four sites in each country. If it is the case that not all facilities were visited, authors should provide justification for the selected facilities. It is unclear why those sites from Kenya, Nigeria, and Uganda were selected.

3. The authors observed biased sampling estimates for several but not all contraceptive method types. In the discussion, authors should provide better explanation on this finding.

4. The authors demonstrated that the likelihood of substantial bias decreases as EA size decreases and as the number of outlets sampled per EA increases. Author are requested to focus more on this result.

5. The household survey will not allow for sampling from all households within the outlet study area. This may thus introduce sampling bias which would cause the household survey data to not be representative of the full study area. The authors should address this issue.

6. The household survey might not designed to follow the same respondent over the 2-3 survey rounds, the study is limited in its ability to explain individual-level changes in FP behavior and any correlation between these changes in behavior and changes in the FP supply market.

7. The survey does not capture the full breadth of patterns of FP use and outlet choices.

8. Most importantly, the study is failed to produce data or findings that are representative at the regional or national levels.

9. Recall issues of family planning may come into play, which is missing in the manuscript.

10. The 2-3 rounds of data collection implemented quarterly may present a burden for outlet respondents, especially those which stock many FP products and brands. Firstly, it may increase the non-response or attrition rate in subsequent rounds. Secondly, it may cause respondents to provide data that are not valid (e.g., junk data in order to finish an interview more quickly). Both of these issues would affect the validity and reliability of the data. Authors should work more on it.

11. There is no statistically predetermined number of outlets that is sought based on a given level of precision. This is not explained in the manuscript.

12. Inclusion and exclusion criteria of the outlet are missing in the manuscript.

6. PLOS authors have the option to publish the peer review history of their article (what does this mean?). If published, this will include your full peer review and any attached files.

Reviewer #1: No

---

## [Author Response · Author response to Decision Letter 0]

2 Jun 2022

REVIEWER’S COMMENTS

Responses to reviewer’s comments

Comments to the Author

General comment:

Authors are advised to elaborate on causes and the impact of different types of bias. In the discussion section, please mention about why only selective contraceptive methods were chosen for biased sampling estimates. Also please elaborate clearly about why likelihood of substantial bias decreases as EA size decreases and when the number of outlets sampled per EA increases. 

RESPONSE: Thank you for these thoughtful comments and questions. We have responded to them in turn below, in blue. 

1. Is the manuscript technically sound, and do the data support the conclusions?

Reviewer #1: Partly

2. Has the statistical analysis been performed appropriately and rigorously? 

Reviewer #1: Yes

3. Have the authors made all data underlying the findings in their manuscript fully available?

Reviewer #1: No

RESPONSE: Having reviewed the PLOS One guidance on patient privacy and exceptions to sharing materials, our project’s policy around potentially identifying geocoordinates, and our pre-existing project data in the public domain, we would like to request an exemption from making the data used for the analysis presented in this manuscript public. These data consist of GPS coordinates for outlets, which when combined with other project data that we have available through our project website (and stored in a public repository), would permit the identification of study participants. 

The decision in CM4FP to make public the HH-outlet distance matrices was a compromise that we took to maximize the availability and usefulness of the project’s geographic data while ensuring that the human subjects involved (in the household surveys, not included in this manuscript) could not be identified from these data. Part of that anonymization included the removal of site location information and attempts to mask outer/inner ring shapes in documentation to avoid the chance of someone being able to superimpose real geographic maps onto regenerated network maps made from the household-outlet distance matrices. The figures presented in our manuscript for PLOS One were a part of that masking effort, as we aimed to render the images unconvertable to real maps of study sites. 

The household-outlet distances are already available in our public data (through www.cm4fp.org), and so our primary concern now is to ensure that any further data release would not increase the likelihood of those data becoming identifying for our female respondents. 

We believe that releasing the geocoordinates used in this manuscript’s analysis, even with jitters (something we have been exploring in response to the PLOS request), when combined with product availability data and quite precise (+/- 10m) outlet-to-household distances, increases the risk of participant identification unacceptably. We therefore hope that you would consider granting an exception to this data sharing requirement in this case.

4. Is the manuscript presented in an intelligible fashion and written in standard English?

Reviewer #1: Yes

RESPONSE: Thank you for these responses. We further address each of the specific comments below. 

5. Review Comments to the Author

Reviewer #1: Review Comments to the Author

The study examined the bias in product availability estimates from contraceptive outlet surveys in Kenya, Nigeria, and Uganda. They found evidence of bias in estimates of contraceptive availability generated from simulated area-based sampling. Within specific study sites and rounds, they observed biased sampling estimates for several but not all contraceptive method types, with bias more likely to occur in sites with heterogeneity in both spatial distributions of outlets and product availability within outlets. I commend the authors for the work done. Authors are encouraged to consider the following issues.

1. The introduction does not highlight the reason for urban focus. Is there a special reason for limiting it urban? Authors could explain the reasons for the selection of geographics in the introduction section.

Can authors also explain bias in product availability estimates from contraceptive outlet in public sectors? It would be good to understand how bias in product availability estimates from contraceptive outlet has been across the countries in the introduction.

RESPONSE: Thank you for this comment. The urban focus of the study reflected the available data from the CM4FP project (which concentrated only on urban/ semi-urban areas with the exception of one site in Uganda). This urban focus was due to the original project objectives of generating supply and demand side data for selected urban and semi-urban sites. More information about the study design has been published elsewhere (https://gatesopenresearch.org/articles/5-176/v1). We have added text to the manuscript (location: page 7) to better reflect the design. 

The private sector is the focus of this paper for a few reasons. First, it is relatively under-researched in supply-side market dynamics studies, and yet is playing an increasingly important role in the provision of FP products and services in the countries included here. Second, we wanted to make a contribution to the literature in relation to the design of similar studies, notably the PMA SDP sampling approach. PMA samples up to three private sector outlets per EA which we were able to simulate using the CM4FP datasets. However, PMA’s public sector sampling approach involves visiting the public sector outlets either within, or near to each EA included, to capture facilities at primary, secondary and tertiary levels of the public sector. The CM4FP census included only those public facilities found within the study areas (not those in neighboring areas), and therefore does not allow for a simulation of the PMA approach. We agree that bias in public sector outlets would certainly be interesting and important to examine, but this was outside of the bounds of the present study. 

There are different causes for bias including area-based bias. Authors should also dedicate a paragraph stating those causes and the impact of different types of bias.

Thanks for this suggestion. We agree that multiple different causes of bias are possible. As the focus of this manuscript is on sampling bias in contraceptive service delivery point estimates, we believe that the key alternative bias (other than area-based sampling bias) is bias that arises due to incomplete ascertainment of facilities on master facility lists (MFLs). (MFLs are used as the sampling frame for traditional, large-scale supply-side environment surveys, including the Demographic and Health Survey’s Service Provision Assessments and the World Health Organizations’ Service Availability and Readiness Assessments.) MFL-based sampling may be subject to sampling bias if specific facilities (such as unregistered private facilities) are less likely to be captured on MFLs. In addition, MFL-based samples may lack generalizability to non-facility outlets that are not captured at all in the sampling frame, such as pharmacies and drug shops. We have added additional text on page 4 to emphasize the impact of this type of bias.

2. The authors stated that for the purpose of the study, they visited hospitals, medical centers, clinics, health centers, pharmacies, and drug shops/chemists/Patent and Proprietary Medicine Vendors (PPMVs), in the study sites. Authors should be clear whether they visited all facilities for four sites in each country. If it is the case that not all facilities were visited, authors should provide justification for the selected facilities. It is unclear why those sites from Kenya, Nigeria, and Uganda were selected.

RESPONSE: Thanks for this comment. All outlets/facilities/vendors within each study site that stocked FP products/ services (more than just male condoms) were included in the study (pages 7 and 9). 

3. The authors observed biased sampling estimates for several but not all contraceptive method types. In the discussion, authors should provide better explanation on this finding.

RESPONSE: Thanks for this question, we have added FP product specific text to the discussion section on page 22. 

4. The authors demonstrated that the likelihood of substantial bias decreases as EA size decreases and as the number of outlets sampled per EA increases. Author are requested to focus more on this result.

RESPONSE: Thanks for this request for clarification. To some extent this result is expected, as increasing sample size per EA, from smaller pools of outlets per EA (i.e. smaller EAs) would result in a closer approximation to a full census of the study area. However, we agree that this finding could be emphasized and have added additional text on page 27 to reiterate this finding to contextualize our recommendations for improving area-based sampling approaches.

5. The household survey will not allow for sampling from all households within the outlet study area. This may thus introduce sampling bias which would cause the household survey data to not be representative of the full study area. The authors should address this issue.

RESPONSE: The present manuscript does not include any data from the Household surveys. These data were simply discussed in the methods section of the manuscript for completeness (as they are a part of the overall CM4FP study). We acknowledge that this has the potential to cause confusion and so have made their exclusion from this analysis clearer in the methods section (page 6). 

6. The household survey might not designed to follow the same respondent over the 2-3 survey rounds, the study is limited in its ability to explain individual-level changes in FP behavior and any correlation between these changes in behavior and changes in the FP supply market.

RESPONSE: As above, the manuscript focusses only on the CM4FP FP outlet data, in which the same outlets were repeatedly surveyed. The analysis does not consider the household data, nor any interactions between supply and demand sides of the FP market. 

7. The survey does not capture the full breadth of patterns of FP use and outlet choices.

RESPONSE: This manuscript focuses exclusively on the FP outlet data, aiming to present novel information about potential bias from samples of outlets. Individual FP user behaviors and preferences are certainly important for our understandings of the FP market, but are beyond the scope of this present analysis. 

8. Most importantly, the study is failed to produce data or findings that are representative at the regional or national levels.

RESPONSE: The CM4FP study as originally conceived was a methodological pilot, aiming to test novel approaches to gathering and analyzing FP supply and demand data. The study’s limitations include not providing representativeness at any geographical level beyond the specific study sites. We have expanded the limitations section to more clearly express this issue (page 23). However, while not representative, we hope that the findings presented in the manuscript are nevertheless generalizable to other studies that employ sampling approaches similar to those simulated here.

9. Recall issues of family planning may come into play, which is missing in the manuscript.

RESPONSE: Thank you for this comment. We agree that recall issues are likely to play a role in FP user datasets. The data that this manuscript relies on is from the outlet censuses however, in which FP products were audited by data collectors. As such, we do not believe that recall would be a significant source of bias in this particular case. 

10. The 2-3 rounds of data collection implemented quarterly may present a burden for outlet respondents, especially those which stock many FP products and brands. Firstly, it may increase the non-response or attrition rate in subsequent rounds. Secondly, it may cause respondents to provide data that are not valid (e.g., junk data in order to finish an interview more quickly). Both of these issues would affect the validity and reliability of the data. Authors should work more on it.

RESPONSE: Thanks for this comment. The FP product data were collected by data collectors (rather than relying on providers to report these data) in order to avoid the burden that the reviewer rightly identifies. We found outlet attrition to be relatively rare (for example, in the Kenya data, of the total 549 unique non-CHW outlets, 478 appeared in all three rounds, and 523 appeared in at least two of the three rounds. Outlets could be lost from the study, or added to the study between rounds. 

11. There is no statistically predetermined number of outlets that is sought based on a given level of precision. This is not explained in the manuscript.

RESPONSE: The CM4FP data on which this manuscript is based consisted of a census of all outlets in a set of geographical areas. The geographical boundaries for the outer ring at each site encompassed one or more contiguous wards (Kenya and Nigeria) or parishes (Uganda), that were completely censused to measure the total market for FP products and services within each ring-fenced area. To determine the geographic boundary of the outer rings, an initial target area was selected to capture a total of 600 outlets across all sites in each country. Because of the exploratory nature of the study and the census approach, the number of outlets included in the study was not statistically predetermined. Rather, the target number of outlets was determined pragmatically to allow for a deep dive into localized family planning markets within the constraints of the available budget and timeframe. Fieldwork was concurrent across study sites within each country, and complete wards or parishes were censused and added to the outer ring sequentially until the target sample size for each country was achieved or exceeded. Once the overall target number of outlets for the country had been achieved, the outlet census was completed in the current ward/parish in each study site, and no further wards/parishes were added. The final number of outlets counted in the census varied across sites, largely due to differences in outlet density within wards or parishes; larger urban area outer rings generally had a greater number of outlets in each ward or parish, and small urban, semi-urban, and rural areas had fewer outlets in each ward or parish.

We have added more detail to the manuscript explaining this approach on page 7

12. Inclusion and exclusion criteria of the outlet are missing in the manuscript.

RESPONSE: Apologies for this omission. Outlets were eligible for inclusion in the census of FP product and service providers if they had stocked at least one modern FP method (aside from male condoms) or offered any FP services during the past three months. Public and private health provider and health retail outlets of all types within the outer ring, including hospitals, health facilities, pharmacies, patent and proprietary medicine vendors (PPMVs), and drug shops, were screened for inclusion. Outlets that served the military but not the general public were excluded, as were general retailers, bars, hotels, and brothels where only condoms are typically available. In the Lagos and Abia sites in Nigeria, a small number of general retailers/supermarkets offered oral contraceptives and/or emergency contraceptives, so these outlet types were screened and included if eligible. In the overall CM4FP study, CHWs were included in the outlet census if they operated in the community as a mobile provider of FP products and not only within brick-and-mortar facilities. Any outlet or CHW from the initial census no longer meeting the inclusion criteria in a subsequent round was excluded from that point forward. However in our analyses, all CHWs were excluded as they did not have a specific geographical location. 

We have added further information to the manuscript on page 9.

---

## [Decision Letter · Decision Letter 1]

11 Jul 2022

Bias in product availability estimates from contraceptive outlet surveys: evidence from the Consumer’s Market for Family Planning (CM4FP) study

PONE-D-21-34612R1

Dear Dr. Bouanchaud,

We’re pleased to inform you that your manuscript has been judged scientifically suitable for publication and will be formally accepted for publication once it meets all outstanding technical requirements.

Kind regards,

Manoj Kumar

Academic Editor

PLOS ONE

Additional Editor Comments (optional):

Reviewers' comments:

Reviewer's Responses to Questions

**Comments to the Author**

1. If the authors have adequately addressed your comments raised in a previous round of review and you feel that this manuscript is now acceptable for publication, you may indicate that here to bypass the “Comments to the Author” section, enter your conflict of interest statement in the “Confidential to Editor” section, and submit your "Accept" recommendation.

Reviewer #1: All comments have been addressed

2. Is the manuscript technically sound, and do the data support the conclusions?

Reviewer #1: Yes

3. Has the statistical analysis been performed appropriately and rigorously? 

Reviewer #1: Yes

4. Have the authors made all data underlying the findings in their manuscript fully available?

Reviewer #1: Yes

5. Is the manuscript presented in an intelligible fashion and written in standard English?

Reviewer #1: Yes

6. Review Comments to the Author

Reviewer #1: All of the comments were addressed. The manuscript looks scientifically sound. No additional comments from the reviewer.

7. PLOS authors have the option to publish the peer review history of their article (what does this mean?). If published, this will include your full peer review and any attached files.

Reviewer #1: **Yes: **Dr. Nitai Roy

---

## [Editor Report · Acceptance letter]

19 Aug 2022

PONE-D-21-34612R1 

Bias in product availability estimates from contraceptive outlet surveys: evidence from the Consumer’s Market for Family Planning (CM4FP) study 

Dear Dr. Bouanchaud:

I'm pleased to inform you that your manuscript has been deemed suitable for publication in PLOS ONE. Congratulations! Your manuscript is now with our production department. 

Kind regards, 

on behalf of

Dr. Manoj Kumar 

Academic Editor

PLOS ONE